# Quantifying the Effects of Biochar Application on Greenhouse Gas Emissions from Agricultural Soils: A Global Meta-Analysis

**Qi Zhang [1,2,3,4], Jing Xiao [5], Jianhui Xue [1,2,*] and Lang Zhang [3,4]**

[1] Co-Innovation Center for Sustainable Forestry in Southern China, Nanjing Forestry University, Nanjing 210037, China; zhangqi1122@126.com

[2] College of Biology and the Environment, Nanjing Forestry University, Nanjing 210037, China

[3] Shanghai Academy of Landscape Architecture Science and Planning, Shanghai 200232, China; zhanglang18@sohu.com

[4] Shanghai Engineering Research Center of Landscaping on Challenging Urban Sites, Shanghai 200232, China

[5] National Engineering Laboratory for Improving Quality of Arable Land, Institute of Agricultural Resources and Regional Planning, Chinese Academy of Agricultural Sciences, Beijing 100081, China; xiao_jing06@163.com

* Correspondence: jhxue@njfu.edu.cn; Tel.: +86-25-84347001; Fax: +86-25-84438177

**Abstract:** Agricultural disturbance has significantly boosted soil greenhouse gas (GHG) emissions such as methane ($CH_4$), carbon dioxide ($CO_2$), and nitrous oxide ($N_2O$). Biochar application is a potential option for regulating soil GHG emissions. However, the effects of biochar application on soil GHG emissions are variable among different environmental conditions. In this study, a dataset based on 129 published papers was used to quantify the effect sizes of biochar application on soil GHG emissions. Overall, biochar application significantly increased soil $CH_4$ and $CO_2$ emissions by an average of 15% and 16% but decreased soil $N_2O$ emissions by an average of 38%. The response ratio of biochar applications on soil GHG emissions was significantly different under various management strategies, biochar characteristics, and soil properties. The relative influence of biochar characteristics differed among soil GHG emissions, with the overall contribution of biochar characteristics to soil GHG emissions ranging from 29% ($N_2O$) to 71% ($CO_2$). Soil pH, the biochar C:N ratio, and the biochar application rate were the most influential variables on soil $CH_4$, $CO_2$, and $N_2O$ emissions, respectively. With biochar application, global warming potential (impact of the emission of different greenhouse gases on their radiative forcing by agricultural practices) and the intensity of greenhouse gas emissions (emission rate of a given pollutant relative to the intensity of a specific activity) significantly decreased, and crop yield greatly increased, with an average response ratio of 23%, 41%, and 21%, respectively. Our findings provide a scientific basis for reducing soil GHG emissions and increasing crop yield through biochar application.

**Keywords:** biochar application; methane; carbon dioxide; nitrous oxide; agricultural soils; meta-analysis

## 1. Introduction

The current global warming trend is of great significance since it is most likely the result of anthropogenic activities, and its control has become a challenge [1]. The global average temperature is predicted to increase by 2 °C by the end of the 21st century [2], and this could be attributed to global warming and atmospheric greenhouse gas (GHG) emissions [3]. Agriculture is one of the main sources of GHG emissions due to human disturbance. Additionally, three major GHGs—methane ($CH_4$),

carbon dioxide ($CO_2$), and nitrous oxide ($N_2O$)—contribute almost 10%–12% of global anthropogenic emissions [2,4]. Meanwhile, the total human population is projected to reach 9.2 million by 2050, which will inevitably require more food and exert huge pressure on agriculture [5].

Biochar application, as a soil amendment, could be an effective strategy for mitigating GHG emissions and increasing crop yield. The response of biochar to soil GHG emissions and crop yield implies that the biochar feedstock characteristics and pyrolysis temperature might contribute to the cost savings of biochar production. Biochar, produced by high-temperature and low-oxygen pyrolysis, is a carbon-rich and charcoal-like product that contains higher recalcitrant carbon; moreover, biochar is considered an environmentally friendly technology [6,7]. However, biochar characteristics and soil properties greatly influence soil GHG emissions and crop productivity [8,9]. Numerous studies examined the positive, negative, and neutral effects of biochar application on soil $CH_4$ and $CO_2$ emissions. For example, biochar application in calcareous and dry cropland significantly reduces soil $CH_4$ emission by 33% compared with soil without biochar [10]. Knoblauch et al. [11] reported that the effect of biochar application on soil $CH_4$ emission is not significant in paddy soil. Wu et al. [12] found no significant effect of biochar application on soil $CO_2$ emission in alkaline soils, whereas a significant 12% decrease was found in the effect sizes in acidic sandy soils. Conversely, Chintala et al. [13] reported a negative effect of biochar application on soil carbon mineralization based on different biomass conversion processes. Many hypotheses were proposed to explain the impact of biochar application on soil GHG emissions. The positive effect of biochar application on soil $CH_4$ and $CO_2$ emissions could be attributed to the increase in belowground primary productivity and the inhibition of soil methanotrophs [14], while reduced soil $CH_4$ and $CO_2$ emissions may be associated with decreased enzymatic activities and the ratio of methanogenic to methanotrophic archaea [15]. Many studies pointed out that soil $N_2O$ emission responds contrarily to biochar application. Increasing soil $N_2O$ emission could be ascribed to improved soil-water content after biochar application, which explains the improvement in soil $N_2O$ emission associated with soil properties [12,16]. Additionally, different raw materials and pyrolysis temperatures used in the production of biochar may also have various effects on soil GHG emissions and crop productivity [17].

Recently, many researchers explored the effects of biochar application on soil GHG emissions by using systematic reviews such as meta-analysis [9,18–20]. Cayuela et al. [21] only emphasized the role of biochar in the mitigation of soil $N_2O$ emission on the basis of 30 studies with 261 treatments. Liu et al. [22] investigated the relationships among soil $CO_2$ emission, soil organic carbon, and microbial biomass carbon and biochar amendment from 50 papers with 395 treatments. Another published meta-analysis showed that biochar application increases $CO_2$ by 22.14% but decreases $N_2O$ fluxes by 30.92%, as well as significantly affecting soil $CH_4$ emission [8,9]. Therefore, it is imperative to explore the simultaneous effects of biochar application on soil GHG emissions, global warming potential (GWP), greenhouse gas emission intensity (GHGI), and crop yield. It is necessary to quantify the major driving factors of soil $CH_4$, $CO_2$, and $N_2O$ emissions from agricultural soils in response to biochar application.

In this study, our objectives were: (1) quantify the response of soil GHG emissions to biochar application under different management strategies, biochar characteristics, and soil properties; (2) explore which environmental variables are the driving factors of the regulation of soil GHG emissions with biochar application; and (3) evaluate the response of soil GWP, GHGI, and crop yield to biochar application under identical conditions.

## 2. Materials and Methods

### 2.1. Data Sources

Data were collected from 129 published articles through November 2019; the focus of the analyzed papers was on the effects of biochar application on soil greenhouse gas (GHG) emissions and crop yield. These selected publications were obtained from the China Knowledge Resource Integrated Database (http://www.cnki.net) and the Web of Science (http://apps.webofknowledge.com). The keywords

included "biochar", "methane", "carbon dioxide", "nitrous oxide" or "greenhouse gases", and "crop yield". To standardize the dataset, five criteria were used: (1) treatments and the control group must be included in the same independent trials; (2) either crop yield or soil GHG emissions must be present in both treatments and control groups; (3) the data of soil GHG emissions must come from topsoil (0–20 cm); (4) the means, standard deviation (or standard error), and number of replicates must be presented or calculated from the reported data; (5) the number of trial design iterations must be higher than two. From the above principles, 204 sets of $CH_4$ data, 194 sets of $CO_2$ data, and 444 sets of $N_2O$ data were finally obtained from the published articles. In addition, 546 sets of crop yield data were also obtained from 129 published articles.

## 2.2. Database Building

Our datasets were collected and placed in Microsoft Excel 2013 software. The complete information in the database included (1) basic information about the experimental conditions: experimental sites (longitude and latitude), experimental duration (month), crop types, and experimental treatments (the application rate and quality of biochar); (2) biochar characteristics: feedstock, pyrolysis temperature, carbon content, nitrogen, carbon-to-nitrogen ratio (C/N), and pH; (3) soil properties: soil texture (the content of clay, silt, and sand), pH, and moisture; (4) target data: crop yield and soil GHG emissions during the experimental period. If some data were reported as graphs, we used the software GetData Graph Digitizer 2.24 to obtain these data indirectly (means and corresponding standard deviation (SD)). If standard error (SE) was given, SD was obtained from SE by the following equation:

$$SD = SE \times \sqrt{n} \tag{1}$$

To investigate the effects of management, biochar characteristics, and soil properties on soil GHG emissions, this dataset was standardized in the first step. Biochar application rates were unified by using the unit of tons (biochar) per hectare (t ha$^{-1}$). For pot experiments, 300,000 kg of soil per acre plow was used to unify the unit of tons (biochar) per hectare (t ha$^{-1}$). Therefore, biochar application rates were divided into four categories: <10 t ha$^{-1}$, 10–40 t ha$^{-1}$, 40–80 t ha$^{-1}$, >80 t ha$^{-1}$); biochar application time was categorized into four groups: <0.5 year, 0.5–1 year, 1–2 years, and >2 years. To assess biochar characteristics, biochar was divided into five categories according to the raw material or feedstock for the meta-analysis: (1) shell residue (nutshell, oat hull, walnut shell, peanut hull, and bagasse); (2) straw waste (peanut straw, corn stalks, wheat straw, sorghum stalks, and rape stalks); (3) wood waste (bark, wood chips, pruning, trunk, and branches); (4) livestock manure (pig manure, cow manure, and sheep manure); (5) municipal solid waste (household waste and excess sludge). For the analysis of the pyrolysis temperature of biochar, four groups of pyrolysis temperature were considered: lower range (≤400 °C), middle range (401–500 °C), middle to high range (501–600 °C), and higher range (≥600 °C). Additionally, biochar characteristics were categorized into five groups according to the carbon-to-nitrogen ratio (C/N) of biochar: <20, 20–50, 50–100, 100–300, and ≥300; biochar pH was divided into five groups: ≤7, 7–8, 8–9, 9–10, and ≥10. For soil properties, soil texture was classified into sandy soil, loamy soil, and clayey soil according to the relative contents of sand (0.05–2 mm), silt (0.002–0.05 mm), and clay (<0.002 mm), respectively. If soil pH was determined from the $CaCl_2$ solution method, then the values of soil pH ($H_2O$) were obtained via the following equation: pH ($H_2O$) = 1.65 + 0.86 pH ($CaCl_2$) [23]. On the basis of soil classification, soil pH was also grouped into acidic soil (pH < 5.5), weak acidic soil (5.5 < pH < 6.5), neutral soil (6.5 < pH < 7.5), and alkaline soil (pH > 7.5).

## 2.3. Meta-Analysis

As a statistical method, meta-analysis systematically integrates multiple independent research results with a common research purpose and then comprehensively evaluates their research results in a quantitative way [24]. The software MetaWin 2.1 was used to calculate the effects of management, biochar characteristics, and soil properties on soil GHG emissions and crop yield. Each set of data

(treatment and control group) was required to contain the mean, SD, and sample size (n). In addition, missing SD values were fitted by the variation coefficient computed for the entire database [25]. The natural log-transformed response ratio (lnRR) was used as an index to describe the effects of biochar application on soil GHG emissions and crop yield. The lnRR value was calculated as

$$\ln RR = \ln(X_B/X_C) \tag{2}$$

where $X_B$ and $X_C$ are the means of the biochar application group and control (nonbiochar application) group, respectively. The variance (V) of X was calculated as

$$V = SD^2{}_B/N_B X^2{}_B + SD^2{}_C/N_C X^2{}_C \tag{3}$$

where $SD_B$ and $SD_C$ are the standard deviation of biochar application groups and nonbiochar application groups, and $N_B$ and $N_C$ are the sample number of biochar application groups and nonbiochar application groups, respectively.

The weighting factor ($W_{ij}$), weighted response ratio ($RR_{++}$), and standard error of $RR_{++}$ (S) were obtained according to the following equations:

$$W_{ij} = 1/V_{lnRR} \tag{4}$$

$$RR_{++} = \frac{\sum_{i=1}^{m} \sum_{j=1}^{ki} W_{ij} RR_{ij}}{\sum_{i=1}^{m} \sum_{j=1}^{ki} W_{ij}} \tag{5}$$

$$S(RR_{++}) = \sqrt{1/\sum_{i=1}^{m} \sum_{j=1}^{ki} W_{ij}} \tag{6}$$

If the 95% confidence interval value of $RR_{++}$ overlaps with zero, then biochar has no effect on soil GHG emissions or crop yield [26]. The weighted response ratio (WSR) can be calculated by using Equation (7):

$$RR = (e^{RR_{++}} - 1) \times 100\% \tag{7}$$

where RR is the weighted response ratio, which is the percentage change (%) under the treatment as compared with the control.

Results from different studies should be tested for heterogeneity. If the *p*-value is more than 0.1, then those results are homogeneous, and the fixed-effect model (FEM) is preferable for meta-analysis. Otherwise, a random-effect model (REM) is considered a better option for nonindependent observations [26]. Global warming potential is the overall impact of the emission of different greenhouse gases on their radiative forcing by agricultural practices. GWP in $CO_2$-C equivalents (kg ha$^{-1}$) was estimated for the emission of different greenhouse gases using the relative radiation effect of forcing factors by following Equation (8) [2]:

$$GWP = 25 \times R_{CH_4} + 298 \times R_{N_2O} + R_{CO_2} \tag{8}$$

where $R_{CH_4}$, $R_{N_2O}$, and $R_{CO_2}$ are the soil $CH_4$, $N_2O$, and $CO_2$ emissions (kg ha$^{-1}$), respectively.

Greenhouse gas emission intensity (GHGI) is the ratio of GWP to crop yield, which can be used to relate agricultural production to soil GHG emissions [27]. GHGI was calculated by following Equation (9):

$$GHGI = \frac{GWP}{crop\ yield} \tag{9}$$

A smaller value of GHGI indicates that a lower GWP is produced to obtain the same crop yield, whereas a larger value implies that producing the same crop yield induces a higher GWP.

For nonindependence, Hungate et al. [19] showed that the treatment of nonindependence is the most important factor in determining the difference in outcomes of similar meta-analyses, and that

strict versus relaxed nonindependence criteria can significantly affect results. Therefore, the inverse of the number of observations per site (referred to as site-weighted) is determined for the control of nonindependence [20]. For example, if we only want to explore the effect of biochar types on soil GHG emissions, then the biochar application rate is nonindependent. However, if we want to explore the effect of biochar application rates on soil GHG emissions, then the biochar application rate must be independent in studies with many observations.

### 2.4. Statistical Analysis

One-way ANOVA was used to examine whether soil greenhouse gas emissions and crop yield differed significantly between biochar application and nonbiochar application. The software MetaWin 2.1 was applied to calculate the effect sizes of management, biochar characteristics, and soil properties on soil GHG emissions. All the graphs were obtained by using the software Origin 8.5. Eight variables (biochar application rate and time, biochar characteristics (type, pyrolysis temperature, carbon-to-nitrogen [C:N] ratio, and pH), and soil properties (soil texture and soil pH)) were retained to calculate the relative influence (%) of biochar application on soil GHG emissions on the basis of a boosted regression tree (BRT) model. Boosted trees were constructed by using the recommended parameter values: learning rate (0.01), bag fraction (0.50), cross-validation (10), and tree-complexity (5) [28]. Because there were continuous numerical variables, the Gaussian distribution of errors was used for all BRT fittings. All BRT analyses were performed with the **gradient boost machinet** (GBM) package in R version 3.3.3. Of the 129 papers, 27% reported the emission of soil $N_2O$, $CH_4$, and $CO_2$; therefore, only this subset of studies was used to calculate net emission impacts such as the global warming potential and global warming intensity.

## 3. Results

### 3.1. Site Distribution and Characteristics of Soil GHG Emissions and Yield

The experimental sites in our study were distributed among 38 countries (Figure 1). These sites were grouped into three climate zones: low-latitude, middle-latitude, and high-latitude climate zones, which ranged from −43.6 to 120.3 in latitude and from −155.7 to 172.5 in longitude. With biochar application, the average values of soil $CH_4$ (0.20 kg C ha$^{-1}$ d$^{-1}$), $CO_2$ (78 kg C ha$^{-1}$ d$^{-1}$) emissions, and crop yield (22 Mg ha$^{-1}$) were higher relative to those without biochar application (Table 1). In contrast, the opposite effect was found for soil $N_2O$ emission compared with nonbiochar application, which decreased by 0.02 kg N ha$^{-1}$ d$^{-1}$.

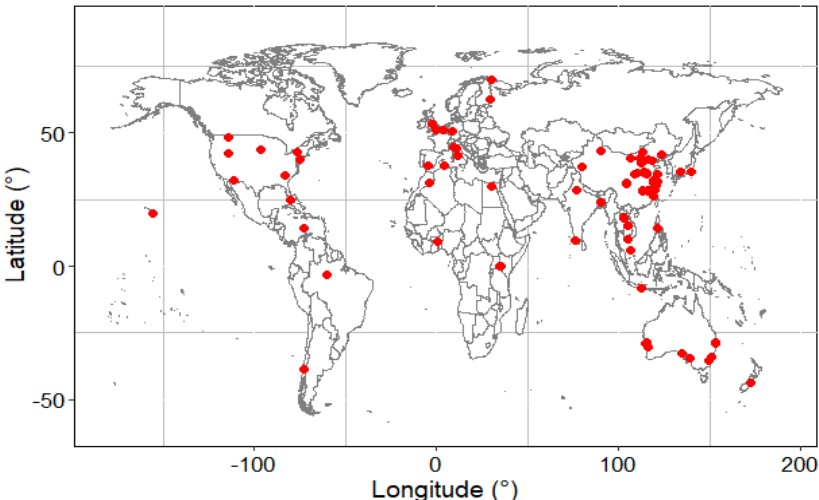

**Figure 1.** Site distribution of studies examining the response of greenhouse gas emissions to biochar application that were included in the meta-analysis.

**Table 1.** Data distribution of methane emission ($CH_4$, kg C $ha^{-1}$ $d^{-1}$), carbon dioxide emission ($CO_2$, kg C $ha^{-1}$ $d^{-1}$), nitrogen dioxide emission ($N_2O$, kg N $ha^{-1}$ $d^{-1}$), and crop yield (Mg $ha^{-1}$) from our datasets.

| | $CH_4$ (kg C $ha^{-1}$ $d^{-1}$) | | $CO_2$ (kg C $ha^{-1}$ $d^{-1}$) | | $N_2O$ (kg N $ha^{-1}$ $d^{-1}$) | | Yield (Mg $ha^{-1}$) | |
| --- | --- | --- | --- | --- | --- | --- | --- | --- |
| | Treatment | CK | Treatment | CK | Treatment | CK | Treatment | CK |
| No. | 204 | 80 | 194 | 79 | 444 | 171 | 564 | 177 |
| Mean | 0.20a | 0.17b | 78a | 65a | 0.04b | 0.06a | 22a | 17b |
| SE | 0.07 | 0.03 | 3.78 | 4.34 | 0.1 | 0.01 | 1.46 | 1.98 |
| Skewness | 3.02 | 4.4 | 3.18 | 3.34 | 1.7 | 2.12 | 4.23 | 4.52 |
| Kurtosis | 10 | 28 | 11 | 12 | 2.43 | 6.32 | 26 | 27 |
| Minimum | −0.05 | −0.07 | 0.01 | 0.01 | 0 | 0 | 0.1 | 0.02 |
| Maximum | 3.89 | 3.65 | 532 | 356 | 0.41 | 0.59 | 477 | 357 |

Note: Treatment, application of biochar; CK, no application of biochar; No., sampling number; SE, standard error. Different letters in the same index are significantly different at $p < 0.05$.

### 3.2. Effects of Management, Biochar Characteristics, and Soil Properties on GHG Emissions

The weighted response ratio of soil GHG emissions exhibited a difference among management strategies, biochar characteristics, and soil properties (Figures 2–4). Across all the sites, compared with nonbiochar application, biochar application significantly increased soil $CH_4$ emission by 15% (95% confidence interval, 2%–27%; $p < 0.05$) and soil $CO_2$ emission by 16% (11%–22%; $p < 0.05$) (Figure 2). Biochar application significantly decreased soil $N_2O$ emission by 38% (27%–45%; $p < 0.05$).

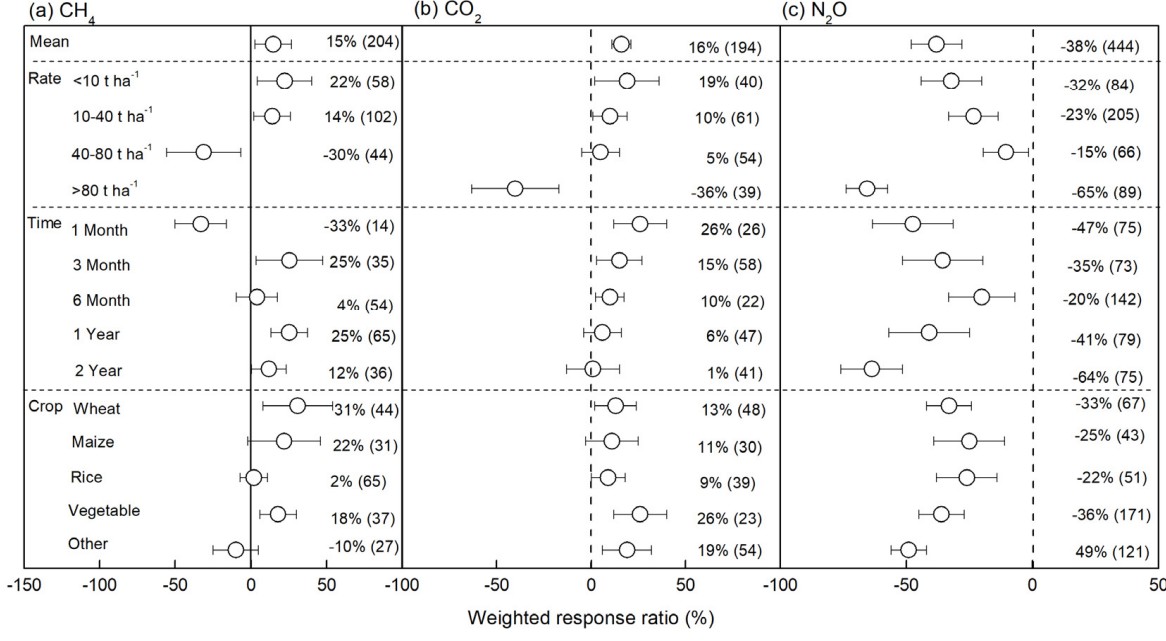

**Figure 2.** The response of (**a**) methane ($CH_4$), (**b**) carbon dioxide ($CO_2$), and (**c**) nitrous oxide ($N_2O$) emissions (%) to the rate and time of biochar application. The numbers in parentheses represent the sample size.

With the increase in biochar application rates, the effect sizes for soil $CH_4$ and $CO_2$ emissions significantly decreased (Figure 2; $p < 0.05$). Biochar application significantly increased soil $CH_4$ and $CO_2$ emissions by 20% and 15% with an application rate of <10 t $ha^{-1}$. Soil $CO_2$ emission shifted from positive to negative (−36%) when the application rate exceeded 80 t $ha^{-1}$. Biochar application appeared to significantly inhibit soil $CH_4$ emission in the first month (−33%). With the increase in experimental time, soil $CO_2$ emission with biochar application decreased from 26% (the first month) to 1% (the second year). Soil $N_2O$ emission was significantly reduced under both the rate and time of

biochar application (15% to 65%; $p < 0.05$). Soil $N_2O$ emission was inhibited and then increased with the increase in the biochar application rate and time. Biochar application significantly increased soil $CH_4$ and $CO_2$ emissions under wheat (31% and 13%) and vegetables (18% and 26%), respectively.

The response of soil GHG emissions to biochar characteristics was significantly heterogeneous (Figure 3). Different feedstock, pyrolysis temperatures, and C/N values of biochar did not significantly affect soil $CH_4$ emission. Biochar with pH < 7, 8–9, and >10 significantly increased soil $CH_4$ emission by 64%, 33%, and 34%, respectively. Biochar from wood waste significantly increased soil $CO_2$ emission (22%). Biochar made at 400–500 °C and 500–600 °C respectively increased soil $CO_2$ emission by 14% and 45%, while biochar made at >600 °C decreased soil $CO_2$ emission by 32%. Biochar with C/N of <20, 20–50, and 100–300 significantly increased soil $CO_2$ emission by 14%, 25%, and 33%, respectively. Biochar with pH < 7, 7–8, and pH > 10 significantly promoted soil $CO_2$ emission by 30%, 67%, and 25%, respectively. Soil $N_2O$ emission could be significantly reduced (11%–61%) by biochar from different feedstock and with different pyrolysis temperatures, C/N values, and pH, except for the following biochar: biochar from shell residue, biochar with a pyrolysis temperature of 500–600, biochar with a C/N value of 20–50, or biochar with pH of >10.

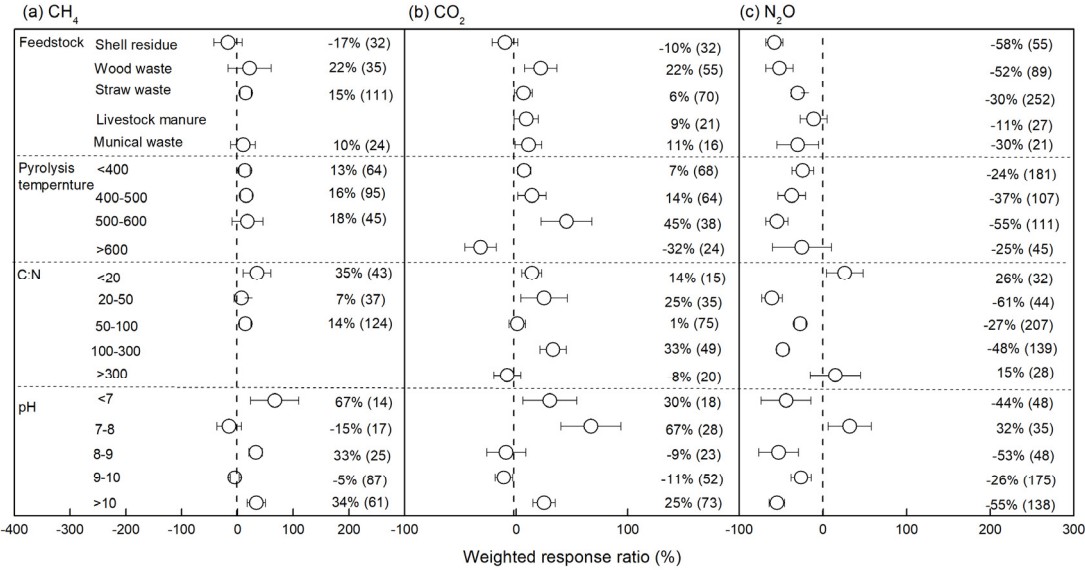

**Figure 3.** Effect of biochar characteristics on (**a**) methane ($CH_4$), (**b**) carbon dioxide ($CO_2$), and (**c**) nitrous oxide ($N_2O$) emissions. The numbers in parentheses represent the sample size.

The increase ratios of soil $CH_4$ emission in loamy soil (10%) and clayey soil (12%) were higher compared with that in sandy soil (5%) (Figure 4). In contrast, soil $CO_2$ emission significantly increased in sandy soil by 25%. The response ratio of soil $CH_4$ emission to biochar input in soil with pH < 5.5 was negative (25%). Biochar application increased soil $CH_4$ emission by 11%, 30%, and 11% in weak acidic soil (5.5 < pH < 6.5), neutral soil (6.5 < pH < 7.5), and alkaline soil (pH > 7.5). Moreover, biochar application significantly increased soil $CO_2$ emission in neutral soil (24%) and alkaline soil (17%). Soil $N_2O$ emission in sandy soil (44%) was higher compared with that in loamy soil (30%) and clayey soil (25%). No significant response to biochar input for soil $N_2O$ emission was found in acidic soil (pH < 5.5). With the increase in soil pH, soil $N_2O$ emission after biochar application decreased from 26% (the first month) to 1% (the second year).

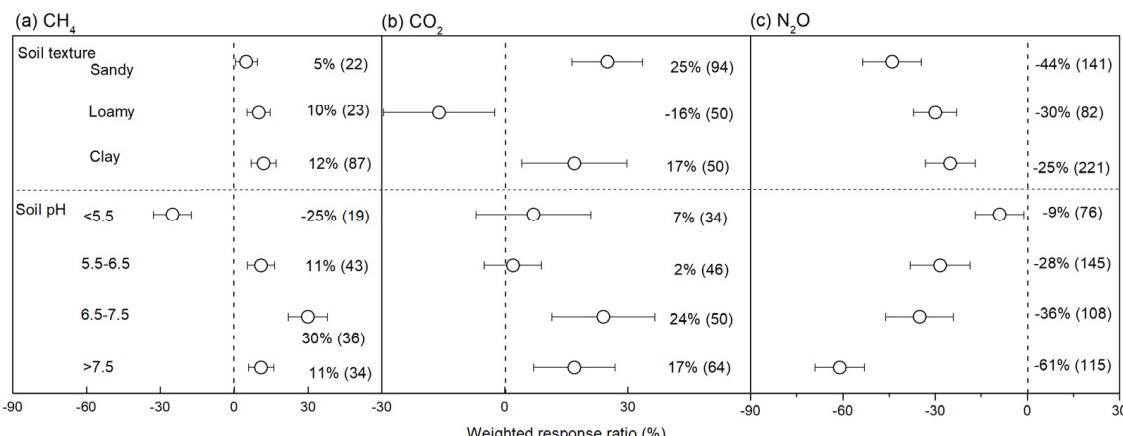

**Figure 4.** The response of (**a**) methane ($CH_4$), (**b**) carbon dioxide ($CO_2$), and (**c**) nitrous oxide ($N_2O$) emissions to the application of biochar under different soil texture and soil pH. The numbers in parentheses represent the sample size.

The results of boosted regression trees showed that the biochar application rate was an influential variable on soil GHG emissions (>15%, Figure 5) among the eight studied variables. The relative individual influence of biochar characteristics was different, and its overall contribution to the observed soil GHG emissions ranged from 29% ($N_2O$) to 71% ($CO_2$). Soil pH had a greater influence on soil $CH_4$ and $N_2O$ emissions (>25%) and less of an effect on soil $CO_2$ emission (<5%). Overall, soil pH, biochar pH, and the biochar C:N ratio were the most influential variables on soil $CH_4$, $CO_2$, and $N_2O$ emissions, respectively.

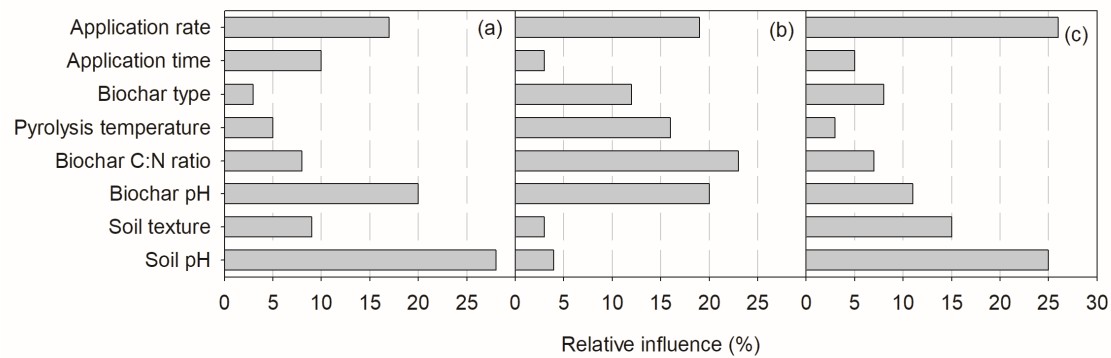

**Figure 5.** The relative influence (%) of predictor variables for the boosted regression tree model of (**a**) methane ($CH_4$), (**b**) carbon dioxide ($CO_2$), and (**c**) nitrous oxide ($N_2O$) emissions. The variables are the rate and time of biochar application, biochar qualities (type, pyrolysis temperature, carbon-to-nitrogen (C:N) ratio, and pH), and soil properties (soil texture and soil pH).

### 3.3. Responses of GWP, GHGI, and Yield to Biochar Application

In general, GWP and GHGI were decreased and crop yield was increased with biochar application in several independent experiments under identical conditions, with an average weighted response ratio of −23%, −41%, and 21%, respectively (Figure 6). Biochar derived from straw waste had a highly remarkable effect: it lowered GWP by 26% and increased crop yield by 3% compared with biochar derived from crop residues. There was no significant decrease in GWP observed in response to the application of biochar from crop residue. However, biochar from straw waste and crop residue increased the crop yield by 18% and 22% and mitigated GHGI by 35% and 48%, respectively.

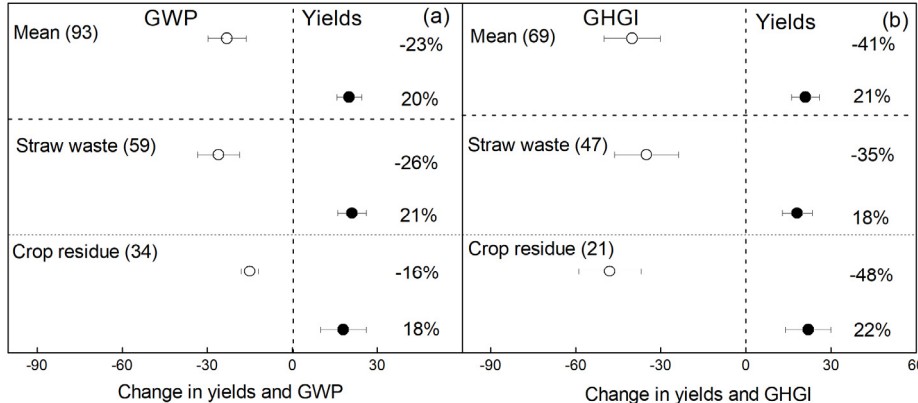

**Figure 6.** The response ratio of crop yield and global warming potential (**a**; GWP) with greenhouse gas emissions intensity (**b**; GHGI) to different biochar types. The numbers in parentheses represent the sample size.

## 4. Discussion

### 4.1. Biochar Effect on Soil GHG Emissions for Varying Management Strategies

The effect sizes of biochar application on soil GHG emissions varied with application rates and experimental years. Our results were consistent with those of Zhang et al. [29], who reported that soil $CH_4$ emission increased when the biochar application rate exceeded 40 t ha$^{-1}$ in paddy soil, but soil $N_2O$ emission was significantly inhibited with the increase in the biochar application rate. Another result from a plain field by Zhang et al. [10] pointed out that soil $CO_2$ emissions increased by 12% with a biochar application rate of 40 t ha$^{-1}$. Our results show that biochar application rates from 10 to 40 t ha$^{-1}$ still promote soil $CO_2$ emission. Soil micro-organisms have been reported to be more likely to decompose water-soluble organic matter in biochar for microbial activity and to release more $CO_2$ in response to a higher biochar application rate [12,15]. Therefore, a biochar application rate of <10 t ha$^{-1}$ is deemed a better choice. Although the short-term effect of biochar application has been discussed in previous studies [30], the long-term benefits of biochar application as a soil amendment are still unclear because the trial lengths were generally 5 years or less [31]. For example, the inhibiting effect of biochar application on soil $CH_4$ emission was reported to occur during the first three months of biochar application [32]. However, this inhibiting effect occurred just in the first month of our analysis. With the increase in biochar application time, soil $CO_2$ emission decreased, but soil $N_2O$ emission first increased and then decreased. This is possibly because biochar application over a long time can destroy soil aggregate structures and release more $CO_2$ from the soil to the atmosphere [33]. In addition, Cayuela et al. [34] claimed that soil $N_2O$ emission decreased as the experimental duration lengthened. Overall, the biochar application rate was an influential variable on soil GHG emissions (>15%, Figure 5). Verhoeven et al. [20] reported that soil $N_2O$ emission reductions were not significantly affected by the cropping system when weighted by an inverse of the number of observations per site. Our results further demonstrate that biochar application significantly increases soil $CH_4$ and $CO_2$ emissions in soil-growing wheat and vegetables.

### 4.2. Biochar Effects on Soil GHG Emissions for Varying Biochar Characteristics

Both the positive and negative effects of biochar on soil GHG emissions were found to be related to biochar characteristics [35]. Consistent with other research findings, our results show that biochar from different raw materials inhibited soil $N_2O$ emission but promoted soil $CH_4$ and $CO_2$ emissions [9,12,14]. Biochar with higher pyrolysis temperatures (501–600 °C) has the potential to reduce soil GHG emissions. Higher pyrolysis temperatures can result in greater surface area, higher ash content, and minimal total surface charge [36]. Aliphatic structure losses at higher pyrolysis temperatures cause the remaining structures to be composed mostly of polycondensed aromatic moieties [36]. Soil micro-organism

activity can be inhibited after the application of biochar with higher pyrolysis temperatures [37]. Biochar with low C/N ($\leq$20) has the potential to promote soil $CH_4$ and $N_2O$ emissions. On the contrary, biochar with a C/N of 20–100 may be the best choice for reducing GHG emissions because the mineralization intensity of soil nitrogen is weakened as the C/N value of biochar increases [38]. Applied biochar can still be mineralized to release nitrogen to the soil, which can stimulate soil respiration and enhance soil $CO_2$ emission. The oxygen-containing functional groups of biochar are alkaline manifestations. Moreover, biochar application increases the cation exchange capacity of soil, which then adsorbs soil $NH_4^+$ and $NO_3^-$ and reduces soil $N_2O$ emission. Therefore, biochar application with high pH inhibits soil $N_2O$ emission. In addition, soil $CH_4$ and $CO_2$ emissions did not correspondingly increase or decrease with the increase in biochar pH. Alkaline biochar application changes soil pH, which diminishes the formation and oxidation of $CH_4$. Therefore, biochar C/N and pH are major factors that affect soil GHG emissions.

### 4.3. Biochar Effect on Soil GHG Emissions for Varying Soil Properties

Soil texture is an important factor through its influence on soil aeration, soil-water content, and nutrient availability. Empirical evidence shows that the effects of biochar on soil GHG emissions and crop growth vary with soil texture [9]. Biochar application in sandy soil and clayey soil increased soil $CH_4$ and $CO_2$ emissions in the analyzed studies. Biochar application in sandy soil is likely to improve soil structure, thus enhancing aeration and permeability for methanotroph communities and increasing $CH_4$ oxidation [24]. Clayey soils are mainly composed of very fine fractions, which are easily bonded to each other because they have greater cementing properties and are occupied by soil water. Therefore, the aeration effect is offset [39]. However, soil $N_2O$ emissions in all soil types were inhibited. Specifically, biochar application could maximize its effect and enhance soil porosity to adsorb $NH_4^+$ while reducing $NO_3^-$ produced by nitrification and $N_2O$ produced by denitrification in sandy soil [40]. Soil pH is regarded as an important factor in soil $N_2O$ emission by ammonia oxidation. Overall, biochar additions to fine-textured soils have a greater effect on soil properties (e.g., soil nutrients, soil micro-organisms, and soil structure), especially soil porosity. Increased porosity exhibited less of an effect on already coarser soils. The effect of biochar in coarse-textured soils has the potential to increase the water-holding capacity, which should, in theory, increase soil $CH_4$ emission. Zhao [41] found soil $NO_3^-$ concentration after biochar application was significantly increased, and soil $N_2O$ emissions were significantly inhibited by laboratory incubation and column leaching studies. Our research also proved a significant decrease in soil $N_2O$ emission (61%; $p < 0.01$) in soil pH > 7.5. The reason for this result might be that an increase in soil pH enhances the activity of nitrifying bacteria's nitrous oxide reductase, which catalyzes more $N_2O$ to $N_2$ and significantly reduces soil $N_2O$ emission [41]. Biochar application increased soil $CH_4$ emission in soil pH > 5.5. In general, soil texture and pH were found to be the most influential variables driving soil $CH_4$ and $N_2O$ emissions with biochar application (Figure 5). In addition, biochar generally has a higher pH than the soil in which it is applied; thus, a liming effect is often induced by biochar application [42]. Biochar's liming effect might also influence the production and consumption of $N_2O$ and $CH_4$. The increase in soil pH induced by the liming effect could create a favorable environment for $N_2O$ reductase and methanotrophic communities, which contribute to the formation of $N_2$ and the oxidation of $CH_4$, respectively [43].

### 4.4. Effects of Biochar Application on GWP, GHGI, and Yield

Global warming potential (GWP) and greenhouse gas emission intensity (GHGI) are simplified indices to estimate the future potential impacts of GHGs on the global climate system [2]. The GHGI was determined by GWP and crop yield. The impact of biochar application on GWP was mainly on soil $N_2O$ emission. This result does not agree with He et al. [9], who reported that biochar application significantly increased GWP by 46% in unfertilized and fertilized soils because fertilization could affect GWP by altering soil $CH_4$, $CO_2$, and $N_2O$ emissions and crop yield.

In addition, biochar from straw residues and crop residues could remarkably decrease GHGI and GWP and increase crop yield at the same time because of the suppression of soil $N_2O$ emission. Biochar application in agricultural soils increases crop yield by enhancing the retention and availability of soil nutrients and water [10,12]. Conversely, Zhang et al. [10] reported that biochar application increases rice yield by 10%–29% at different biochar application rates. Overall, our results show that biochar application significantly decreased GWP by 23% while significantly increasing crop yield by 20% (Figure 6a). Therefore, a significant reduction in yield-scaled GHGI (41%) was detected after biochar application (Figure 6b). Biochar plays a significant role in the mitigation of soil $N_2O$ and $CH_4$ emissions and improvement in crop productivity [24]. A number of mechanisms have been proposed to explain how biochar influences the production and consumption of $N_2O$ in soils: (1) the biochar liming effect leads to an increase in soil pH; (2) enhanced soil aeration restrains denitrification as more $O_2$ is present in soils; (3) the adsorption of $NH_4^+$ and $NO_3^-$ by biochar decreases the substrate availability for nitrification and denitrification; and (4) inhibitory or toxic compounds of biochar are released into soils and inhibit nitrification or denitrification [27]. There are several potential mechanisms by which biochar decreases soil $CH_4$ emission from soils: (1) biochar improves soil aeration, which may stimulate $CH_4$ oxidation and/or suppress $CH_4$ production; (2) biochar has the capacity to adsorb $CH_4$ on its surface; and (3) biochar increases methanotrophic abundances and decreases the ratios of methanogenic to methanotrophic abundances under anoxic conditions [9,22]. Therefore, our study shows that every type of biochar application had a consistent and positive effect on crop yield.

Although the short-term effect of biochar on soil GHG emissions and crop yield was analyzed, the sustainability of biochar for long-term application needs further research. Long-term trials, particularly under field conditions, are required to investigate the impact of biochar on reducing GHGI.

## 5. Conclusions

The effect of biochar amendment on soil GHG emissions varied with the application rate of biochar, biochar characteristics, and soil conditions. Biochar application enhanced the soil $CH_4$ and $CO_2$ emissions but reduced the $N_2O$ flux. With biochar application, the global warming potential and greenhouse gas emission intensity decreased and crop yield increased, indicating that biochar might be an effective amendment for mitigating soil GHG emissions and increasing crop yield. However, soil pH, biochar C:N ratio, and biochar application rate were the most influential variables on $CH_4$, $CO_2$, and $N_2O$ emissions. Furthermore, the response ratios of soil GHG emissions to biochar application were significantly heterogeneous among soil-management strategies, characteristics of biochar, and soil properties. The large uncertainty across the experiments with different lengths of duration constrained the robust characterization of the possible mechanisms through which biochar affected GHG emissions. Thus, well-designed long-term field experiments are urgently needed for an increased understanding of microbial and C dynamics with biochar in agricultural soils.

**Author Contributions:** This research was designed and written mainly by Q.Z. and L.Z.; J.X. (Jing Xiao) provided the dataset; J.X. (Jianhui Xue) contributed methodological and conceptual ideas. All authors have read and agreed to the published version of the manuscript.

**Funding:** This work was supported by the Special Fund for Scientific Research of Shanghai Landscaping & City Appearance Administrative Bureau (grant number G160202), the Major State Basic Research Development Program of China (grant number 2016YFC0502605), and the Science and Technology Program of the Ministry of Housing and Urban-Rural Construction (grant number 2018-K6-001).

**Conflicts of Interest:** The authors declare no conflict of interest.

## Acronyms and Abbreviations

| | |
|---|---|
| GHG | greenhouse gas |
| $CH_4$ | methane |
| $N_2O$ | nitrous oxide |
| $CO_2$ | carbon dioxide |
| GWP | global warming potential |

GHGI        greenhouse gas emission intensity

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
