# Peer review of "Quantifying the Effects of Biochar Application on Greenhouse Gas Emissions from Agricultural Soils: A Global Meta-Analysis"

_sustainability, doi:10.3390/su12083436_

Round 1

Reviewer 1 Report

I dont have further comments. It is ready for publication.

Author Response

Q: I dont have further comments. It is ready for publication.

Re: Thanks for your help.

Reviewer 2 Report

Dear Authors

Most improvements have been made.

The English is not understandable in some of the parts added based on our suggestions.  The sentences just do not make sense. For example the last few sentences of the discussion. The text in red are my comments

" Although the effect of biochar on soil GHG emissions and crop yield was discussed, the sustainability of biochar for long-term application call for a deeper understanding of the process. [the first part of this sentence does not relate to the second part. Adding the word "short term" to the first part will improve the sentence. In the second part instead  "call for a deeper understanding of the process" can be replaced by "need further research."]  Therefore, [wrong word here leave it out] long-term trials, particularly under field conditions, are required to clarify [use investigate instead of clarify] the impact of biochar on reducing GHGI. On the other hand , plant material used as a feedstock for biochar production is a sustainable approach [this is a completely new topic that does not relate to the previous part in the paragraph. One need tosay first that there is not sufficient biochar for use worldwide and one can say that using crop residue for biochar should be examined. There is no proof that this is a sustainable approach!} we need to further address the sources, methods and production cost of biochar before widespread application of biochar can be recommended. Furthermore, the cost per unit reduction of GHG would be a valuable metric together with the yield scaled-GHGI and GWP. [again this does not relate to the previous sentence and needs context]

The manuscript is attached with highlighted in yellow the text that needs to be improved

Author Response

Reviewer 2:

Q1: The English is not understandable in some of the parts added based on our suggestions. The sentences just do not make sense.

Re: Thank you for your comments. We have asked a professional editing service (https://www.aje.com/) to edit the English language and correct any grammar errors.

Q2: Line 30-32 Meaning of these two sentences do not ake a lot of sence. It need to be clearly stated that there is problem that the amount of biochar in the world is limited for mitigating climate change.

Re: These sentences have been revised as "Our findings provide a scientific basis for reducing soil GHG emissions and increasing crop yield by biochar application".

Q3: Line 123 "...forced on..." Wrong word!

Re: It has been revised as "focus on".

Q4: Line 355 Figure title not correct.

Re: It has been revised as "Site distribution of studies examining the response of greenhouse gas emissions to biochar application that were included in the meta-analysis".

Q5: Line 356-358 all should be singular I believe.

Re: Thank you for your comments. It was singular and has been reviewed in all manuscript.

Q6: Line 359 "Yields (Mg ha-1)" yield should be not plural.

Re: It has been revised as "yield" in all manuscript.

Q7: Line 374 "Note Treatment...biochar." something incomplete.

Re: "No., sampling number, SE; standard error" has been added.

Q8: Line 588-591 the English is poor and frankly i do not understand it..

Re: These sentences have been revised as "Although the short-term effect of biochar application has been discussed in previous studies, the long-term benefits of biochar application as a soil amendment are still unclear because the trial lengths were generally within 5 years".

Q9: Line 717-796 this ir poorly worded and this is not the way to end an manuscript.

Re: We have revised in below questions (Q10-13).

Q10: "Although the effect of biochar on soil GHG emissions and crop yield was discussed, the sustainability of biochar for long-term application call for a deeper understanding of the process. [the first part of this sentence does not relate to the second part. Adding the word "short term" to the first part will improve the sentence. In the second part instead  "call for a deeper understanding of the process" can be replaced by "need further research."] 

Re: The sentence has been revised as "Although the short-term effect of biochar on soil GHG emissions and crop yield was discussed, the sustainability of biochar for long-term application needs further research".

Q11: Therefore, [wrong word here leave it out] long-term trials, particularly under field conditions, are required to clarify [use investigate instead of clarify] the impact of biochar on reducing GHGI.

Re: The sentence has been revised as "long-term trials, particularly under field conditions, are required to investigate the impact of biochar on reducing GHGI".

Q12: On the other hand, plant material used as a feedstock for biochar production is a sustainable approach [this is a completely new topic that does not relate to the previous part in the paragraph. One need tosay first that there is not sufficient biochar for use worldwide and one can say that using crop residue for biochar should be examined. There is no proof that this is a sustainable approach!} 

Re: We deleted the sentence as your right comment.

Q13: we need to further address the sources, methods and production cost of biochar before widespread application of biochar can be recommended. Furthermore, the cost per unit reduction of GHG would be a valuable metric together with the yield scaled-GHGI and GWP. [again this does not relate to the previous sentence and needs context]

Re: We deleted the sentence as your right comment.

Q14: The manuscript is attached with highlighted in yellow the text that needs to be improved

Re: Thank you again for your comments and suggestions. We have revised directly these comments in the manuscript and answered the above question (Q1-9).

This manuscript is a resubmission of an earlier submission. The following is a list of the peer review reports and author responses from that submission.

Round 1

Reviewer 1 Report

Dear auhtors, look carefully to the comments provided in the manuscript.

1-The numbers of the greenhouse formula should go in subscript. Fix them.

2-The degree symbol for Celsius doesnt have a line below the symbol.

3-Use three significant figures for the numbers i.e. 26.37- 26.4

4. You have several references in the script with all the last names of the authors, use et al. [x].

5. Line 94- change word putted for placed. Always use scientifically sound words. Same for word "dire" in line 70.

6- Not clear the unit hm-2 if you are saying you are using hectares. ( Hectare symbol is ha. above you mentioned hectare and square meters, to different units.

7- Use of to many 'That" in places no needed. Revise your english..

8- EMPHASIZE THE NOVEL PART OF YOUR STUDY. 

9- vague discussion and conclusiom

7-Line 117-118-use the correct symbol for commas

8- line 120- add spece between numbers and units

9- The figure titles should go below figures. In most of your figures, the title is left in the previous page. Make sure they move together. Also, if there was statistical analysis, include it in the title as a second sentence.

10. Figures 2-4, the Greenhouse formulas needs to be bigger. Make sure when you print the manuscript is readible. If the units is percentage, it should be included in the title.

11. Also, the figure should be above the explainatory parragraph.

12. For figure 4- you are using coarse, medium and fine material and you talk about clay, sand and silt content. Both should be define before to be used. Try to be consistent.  The information should be added in materials and methods and be consistent with the use of them in the figures and results and discussion section.

Author Response

Reviewer 1

Q: Dear authors, look carefully to the comments provided in the manuscript.

Re: Thank you for your comments. We have made a major revision according to your comments, which are really helpful to improve the quality and science of this manuscript.

Q: 1-The numbers of the greenhouse formula should go in subscript. Fix them.

Re: Done as a suggestion.

Q: 2-The degree symbol for Celsius doesn’t have a line below the symbol.

Re: We have revised as 2 ˚C. Thanks

Q: 3-Use three significant figures for the numbers i.e. 26.37- 26.4

Re: We have revised throughout the manuscript.

Q: 4. You have several references in the script with all the last names of the authors, use et al. [x].

Re: Changes have been made accordingly as suggested. Thanks.

Q: 5. Line 94- change word putted for placed. Always use scientifically sound words. Same for word "dire" in line 70.

Re: We have revised as your suggestion.

Q: 6- Not clear the unit hm-2 if you are saying you are using hectares. ( Hectare symbol is ha. above you mentioned hectare and square meters, to different units.

Re: We have revised as ha-1 throughout the manuscript.

Q: 7- Use of to many 'That" in places no needed. Revise your English.

Re: It has been checked, and carefully corrected throughout the manuscript.

Q: 8- Emphasize the novel part of your study. 9- vague discussion and conclusion

Re: We have added the discussion and rewrote the conclusion.

Q: 7-Line 117-118-use the correct symbol for commas

Re: Done as suggested.

Q: 8- line 120- add space between numbers and units

Re: Done as suggested.

Q: 9- The figure titles should go below figures. In most of your figures, the title is left in the previous page. Make sure they move together. Also, if there was statistical analysis, include it in the title as a second sentence.

Re: It has been revised as suggested. Thanks. 

Q: 10. Figures 2-4, the Greenhouse formulas need to be bigger. Make sure when you print the manuscript is readable. If the units are percentage, it should be included in the title.

Re: We have carefully modified figures and revised it. Q: 11. Also, the figure should be above the explanatory paragraph.

Re: Done as suggested.

Q: 12. For figure 4- you are using coarse, medium and fine material and you talk about clay, sand and silt content. Both should be defining before to be used. Try to be consistent.  The information should be added in materials and methods and be consistent with the use of them in the figures and results and discussion section.

Re: we have implemented changes as suggested. Thanks. 

Reviewer 2 Report

Review: Sustainability 676762

Title: Quantification for the effects of biochar application on greenhouse gas emission from agricultural soils: A global meta analysis

The dataset needs to be updated (2016 last date of collection is now 3 years old), a VERY similar meta analysis was published in 2017 (He et al. 2017); the data structure is unclear (i.e. presented as a comparison of GHGs, but the same studies or sets of studies do not all appear to include all the data, CO2, CH4, N2O, yield); cropping system and field versus greenhouse study differences are not discussed (i.e. paddy vs upland has been found to be important, Verhoeven et al. 2017); English grammar needs thorough editing; non-independence of observations coming from a single trial needs to be accounted for/discussed (i.e. multiple biochar types or rates relative to one control in one system/environment are not independent).  Overall the discussion appears weak and presents vague mechanisms of biochar activity that can justify results one way or another.  In a revision with updated data, I would focus on relative influence analysis, this is most novel aspect of the study.

Some specific examples are given below:

L19-20.  This sentence is an example of what I believe is a misleading statement.  A different set of studies (n varies from 132 to 318) was used for each of these categories and this should be expressly stated.  The emissions of CO2, CH4 and N2O can only be compared to each from studies where all three were measured. 

L64.  A number or more recent meta analyses have been conducted on this topic: Cayuela et al. 2015, Verhoeven et al., 2017, He et al. 2017.  References and dataset MUST be updated or a novel take on the analysis needs to be included.

L67.  Check citation style – long list of authors like this seems incorrect.

L143.  A weighting mechanism for non-independence of observations should be considered.  See Hungate et al. 2009 and Verhoeven et al. 2017.

L153.  How many studies and observations reported all 3 gases + yield?  Strongly recommend limiting scope of meta-analysis to these studies, then comparisons across gases can be made.

L189.  Higher than what?  This sentence needs something in it for comparison.

L200.  Value should be <10 t hm-2 not >

L299. Example of discussion that could be improved.  It could equally, if not more logically be rationalized that biochar addition to fine textured soils would have a greater effect on soil structure and porosity and increase porosity and would have a lesser effect on already coarser soils.  Indeed, I believe this is what has been mostly found.  The effect in coarser soils is more likely one of increased water holding capacity, which should in theory increase CH4 emissions.  Mechanistic studies should be cited and a more comprehensive analysis of possible findings here are needed.

Cayuela, M.L., Jeffery, S. and van Zwieten, L., 2015. The molar H: Corg ratio of biochar is a key factor in mitigating N2O emissions from soil. Agriculture, Ecosystems & Environment202, pp.135-138.

He, Y., Zhou, X., Jiang, L., Li, M., Du, Z., Zhou, G., Shao, J., Wang, X., Xu, Z., Hosseini Bai, S. and Wallace, H., 2017. Effects of biochar application on soil greenhouse gas fluxes: a meta‐analysis. Gcb Bioenergy9(4), pp.743-755.

Hungate, B.A., Van GROENIGEN, K.J., Six, J., Jastrow, J.D., Luo, Y., De GRAAFF, M.A., van Kessel, C. and Osenberg, C.W., 2009. Assessing the effect of elevated carbon dioxide on soil carbon: a comparison of four meta‐analyses. Global Change Biology15(8), pp.2020-2034.

Verhoeven, E., Pereira, E., Decock, C., Suddick, E., Angst, T. and Six, J., 2017. Toward a better assessment of biochar–nitrous oxide mitigation potential at the field scale. Journal of environmental quality46(2), pp.237-246.

Author Response

Q: The dataset needs to be updated (2016 last date of collection is now 3 years old), a VERY similar meta analysis was published in 2017 (He et al. 2017); the data structure is unclear (i.e. presented as a comparison of GHGs, but the same studies or sets of studies do not all appear to include all the data, CO2, CH4, N2O, yield); cropping system and field versus greenhouse study differences are not discussed (i.e. paddy vs upland has been found to be important, Verhoeven et al. 2017); English grammar needs thorough editing; non-independence of observations coming from a single trial needs to be accounted for/discussed (i.e. multiple biochar types or rates relative to one control in one system/environment are not independent).  Overall the discussion appears weak and presents vague mechanisms of biochar activity that can justify results one way or another.  In a revision with updated data, I would focus on relative influence analysis, this is most novel aspect of the study.

Re: Thank you for your comments. This paper is innovative in factor quantization, which is different from other studies. We didn't update the database because of enough data and 10 days to revise. You are right. Only a fifth of the data reported these variables at the same time. We have added a discussion about the cropping system. We only give an overall answer to the effects of biochar on GHG and crop yields under different conditions. Therefore, the non-independence was not considered due to the limited amount of data. Only a fifth of the data reported these variables at the same time. We have added uncertainty analysis in DiscussionEnglish has been checked, and carefully corrected throughout the manuscript. We have added some mechanisms in Discussion.

Q1: L19-20.  This sentence is an example of what I believe is a misleading statement.  A different set of studies (n varies from 132 to 318) was used for each of these categories and this should be expressly stated.  The emissions of CO2, CH4 and N2O can only be compared to each from studies where all three were measured. 

Re: This sentence was revised as "Overall, biochar application significantly increased soil CH4 and CO2 emissions by averaged 10% and 12% but decreased soil N2O emission by averaged 36% based on the 87 published papers".

Q2: L64.  A number or more recent meta-analyses have been conducted on this topic: Cayuela et al. 2015, Verhoeven et al., 2017, He et al. 2017.  References and dataset MUST be updated or a novel take on the analysis needs to be included.

Re: We have cited these new references and emphasized our innovation "quantify the major driving factors for soil CH4, CO2, and N2O emissions in response to biochar application from agricultural soils".

Q3: L67.  Check citation style – long list of authors like this seems incorrect.

Re: We have revised the sentence and style of reference citation according to the journal. Thanks

Q4: L143.  A weighting mechanism for non-independence of observations should be considered.  See Hungate et al. 2009 and Verhoeven et al. 2017.

Re: Thanks for your suggestion. We only give an overall answer to the effects of biochar on GHG and crop yields under different conditions. Only a fifth of the data reported these variables at the same time. Therefore, the non-independence was not considered due to the limited amount of data. We have added uncertainty analysis in Discussion.

Q5: L153.  How many studies and observations reported all 3 gases + yield?  Strongly recommend limiting scope of meta-analysis to these studies, then comparisons across gases can be made.

Re: Thanks for your suggestion. That's what we planned to do at the beginning. But, only a fifth of the data reported these variables at the same time.

Q6: L189.  Higher than what?  This sentence needs something in it for comparison.

Re: Done as "difference".

Q7: L200.  Value should be <10 t hm-2 not >

Re: Done as "t ha-1".

Q8: L299. Example of discussion that could be improved.  It could equally, if not more logically be rationalized that biochar addition to fine textured soils would have a greater effect on soil structure and porosity and increase porosity and would have a lesser effect on already coarser soils.  Indeed, I believe this is what has been mostly found.  The effect in coarser soils is more likely one of increased water holding capacity, which should in theory increase CH4 emissions. Mechanistic studies should be cited and a more comprehensive analysis of possible findings here are needed.

Re: We have added some mechanisms in Discussion, which included your view. Due to professional limitations, we finally decided not to refer to mechanistic studies, although it was a good view. Thanks for your good suggestions, again.

Reviewer 3 Report

Dear authors

Please see attached pdf

Interesting article

The paper is generally well written and interesting, however, the negative aspects of biochar need to be addressed (i.e., the damage done to the environment by producing biochar) before the authors can conclude that biochar application will be significant in reducing future temperature increases.

Regards

Author Response

Q: The manuscript gives an interesting overview of biochar experiments that have been carried out all over the world. The conclusions that biochar is a remedy against climate change cannot be made based on this review as detailed below in the comments.

Re: Thank you for your comments. We have made a major revision according to your comments, which are really helpful to improve the quality and science of this manuscript.

Q1: Throughout the manuscript the digits in the data reported should be limited to the significant digits only. For example, the third digit in the percentages is meaningless in the following example from the abstract an “The relative individual influence of biochar characteristics was different on soil GHG emissions, but the overall contribution of biochar characteristics to observed soil GHG emissions was from 32.7% (N2O) to 69.9% (CO2). Soil pH, biochar pH, and biochar application rate were the………”

Re: Thank you very much for good suggestion. We have followed the rules for significant figures and corrected accordingly.

Q2: The following sentence in the abstract is difficult to understand since “global warming potential” definition is not known by most readers. They certainly have no idea what an “average weighted response ratio” “Global warming potential and intensity of greenhouse gas emission significantly decreased, and crop yields greatly increased by biochar application in identical conditions, with average weighted response ratio of -21.6%, 36.8%, and 18.7%, respectively.”

Re: We have explained all these terms in the methodology section.

Q3: Line 28: “Our findings provided a theoretical basis for rational biochar application to soil improvement and climate change mitigation.” This paper is a literature review and not a theoretical analysis and moreover there is nothing rational about the findings on soil improvement and climate change mitigation since this literature review is based on small scale experiments upscaled to the world without considerations about the practicability and the cost of applying biochar to all agricultural land in the world.

Re: Yes, we agree with you. We have revised the conclusion statement in the abstract as “Evidence suggested large-scale biochar application in soils can contribute to mitigating climate change via sequestering carbon in the soil. Further, the practicability of biochar as a climate change mitigation approach requires its effect on GHG budgets over both short and long time scales”

Q4: Around line 42 it is mentioned without a reference that “Biochar application is a sustainable approach to regulate the global emission of soil GHGs and to increase crop yield as well.” Indeed, this manuscript shows that biochar can reduce emissions of GHG’s and increase crop yield based on the publishes literatures. It is highly likely that the published and refereed literature is biased upward because findings that biochar has not any effect are much more difficult to publish. A much greater problem with this statement is the question if this practice is “sustainable”. Biochar originates from plant material. If this is a sustainable practice, the authors need to address the origin of the biochar and the method and the cost to produce it. I might be biased because in writing a proposal about a proposal about application of biochar to agricultural land on a large scale, the biochar produced pulled out the proposal, because he could find a much better price for the biochar, he produced for purposes other than biochar. If the authors want both to upscale their results of the small-scale experiments that they reviewed to the world scale and make definite statement that biochar will prevent climate change, the question of availability and fate of the land where the organic matter is produced for the biochar should be evaluated

Re: Thank you very much for the comprehension overview of the statement. We have changed the sentence to make it clear and avoid ambiguity. Now, the new sentence is added as “Biochar application is an effective strategy to eliminate the global emission of GHG and to increase crop yield (Liu et al., 2019). However, the response of biochar to GHG emission and crop yield implies the characteristic of biochar feedstock, and pyrolysis temperature might contribute to saving the cost of biochar production.” For the second suggestion, we think this is a good and meaningful suggestion, and we have been trying to answer. As you said, we did this manuscript through data collation, so it is difficult to quantify this question. We have added a related discussion in the Discussion.

Q5: Around line 63 “Recently, many researchers explored the effects of biochar application on soil GHG emissions based on huge data integration and analysis” What is “on huge data integration”?

Re: We have revised the sentence for readability as “Recently, many researchers explored the effects of biochar application on soil GHG emissions based on systematic reviews such as meta-analysis”

Q6: Line 68: “Song, Pan, Zhang, Zhang and Wang [8] and He, Zhou, Jiang, Li, Du, Zhou, Shao, Wang, Xu, Hosseini Bai, Wallace and Xu [9] obtained a central tendency of soil GHG emissions in responses to biochar application with 61 and 91 studies. There must a better way than writing out all the authors

Re: We have revised the sentence and style of reference citation. Thanks

Q7: Line 71: In this study, our objective were to (1) quantify the response of soil GHG emissions to biochar application under different managements, biochar characteristics, and soil properties; (2) explore which environmental variables are the driving factors to regulate soil GHG emissions under biochar application; and (3) evaluate the response of soil GWP, GHGI, and crop yields to biochar application under identical conditions. The abbreviation in the objectives forces the reader the find to look through the whole introduction to find the meaning. It would be much better to write the abbreviations out in the parts of the manuscript that are essential for understanding the research in the paper when a reader is scanning the paper

Re: we have implemented changes as suggested and have written the acronyms and abbreviations before the introduction section.

Q8: Line 150 “Global warming potential (GWP) and greenhouse gas emission intensity (GHGI) were calculated from the following equations: …….. (8) GHGI = GWP/Output (9) GWP in the fourth formula represented the relative radiation effect of a given substance compared” The GWP and other variables in Eq 8 have units which are important because there are constants in the equation that are not dimensionless. What is output in eq 9? What is the “fourth” in the last sentence?

Re: According to EPA and IPCC, GWP used as a factor of GHG emission instead of presenting their respective units. The output is crop yield and is explained in Materials and Methods. It has been rephrased as equation (8) and (9).

Q9: Line 175 “The experimental sites of our study were distributed in 26 countries (Fig. 1). These sites were grouped into three climate zones: low latitude, middle latitude and high latitude climate zone, which covered from -43.56 to 62.50 in latitude and from -155.70 to 172.46 in longitude. The average values of soil CH4 (0.19 kg C/ ha/d) and CO2 (69.83 kg C/ ha/d) emissions and crop yields (20.66 106 g ha/ha/d ….” Watch the significant digits! Check Table 1 as well (and the rest of the manuscript) In n addition in metric 106 g is written as Mg

Re: It has been revised as suggested. Thanks. 

Q10: Figure titles should be below the figures and not above (Fig 2 etc. ).

Re: It has been modified.

Q11: In the text the meaning of weighted response ration should be explained in layman’s terms. Not clear to me after looking at the equation.

Re: It has been added in the methodology section.

Q12: In all figures digits behind the decimal point should be removed

Re: Done as suggested. Thanks

Q13: In the discussion the question needs to be addressed on the availability of Biochar to upscale the results. In addition, the cost per unit of greenhouse gas reduction would be a useful metric together with the global warming potential. Re: Please see the reply to Q4.

Round 2

Reviewer 2 Report

Comments:

Overall: English grammar still needs editing. Many instances where the tense is off, for example in the introduction, “the total human population would be reached to 9.2 million… ”

Up to the editor to extend a revision timeline, I still feel that dataset should be updated before a revision.

Carefully review revisions, in some cases the grammar and sentence do not make sense. It appears that sentences were inserted in response to a reviewer without much consideration. Example, L761-762 or L571.

Specific comments

L18: Add in a sentence that clearly states the data structure and will reduce mis-interpretation.

“Of the 87 papers, 20% (or one fifth) reported emissions of N2O, CH4 and CO2; therefore only this subset of studies was used to calculate net emission impacts such as the global warming potential and global warming intensity.”   

L19-20: “by averaged” not correct English. Change to “by on average”

L28-30:  Revise or eliminate this sentence. This meta-analysis does not examine carbon sequestration in the soil!

L74: Biochar application, nor any practice will eliminate GHG.  Please revise.

L79: change ‘profoundly’ to ‘can’.  Profoundly is a quite a dramatic statement.

L102: State the actual finding of this study, not simply repeat ‘obtain a central tendency’ – that is what they did, not the results.  State, 22% increase in CO2, 31% decrease in N2O and no effect on CH4. Also, He et al. 2017 had 91 papers, was their criteria less strict?  More up to date?  

L210: I am not sure the authors understand correctly potential non-independence in this dataset.  I do not mean non-independence between CO2/CH4/N2O.  Rather, most studies report a control versus multiple types or rates of biochars.  Some studies may have 5+ observations, all against one control, in one climate, one soil, etc.  Observations from the same study compared to one control are not independent of each other.  This can create real bias.  See Verhoeven et al. 2017 and Hungate et al. 2009. Bias towards results for small plot and pot studies is more likely because these tend to have more treatments (i.e. observations), while larger, longer term field studies – those that are actually more reflective of real conditions – generally have fewer treatments. Adding a weighting mechanism to account for the number of observations from a given study can help remedy this.

L571:  Did you examine results by cropping system?  Where is this analysis? I suggested adding this analysis, but I do not think this was done. For example, look at effects in cereal versus vegetable vs perennial cropping systems; rice versus upland…

L635: Was the type of clay reported in each study?  I doubt it.  If you are referring to general properties of clays and clayey soils, please revise the sentence.

L637: Really?  This seems like a big finding.  Was N2O actually inhibited, seems speculative.  This was not measured.  If indeed there was a negative correlation between N2O and CO2 and CH4, I would show that with a graph or r2 and state ‘negative correlation’. 

L653-656: Combine discussion on liming with mention of liming on L644/645.

L773: After “… but inhibited N2O emissions” add “, all three gases were only measured in X% of studies.”

L779: “characterization of microbial changes with biochar” .  Please revise, no attempt to characterize microbial pathways was made and cannot be made with the current dataset.  Change to something like, “… robust characterization of possible mechanisms through which biochar affected GHG emissions”

Reviewer 3 Report

Please see attached pdf
